# Candelilla Wax Edible Coating with *Flourensia cernua* Bioactives to Prolong the Quality of Tomato Fruits

**DOI:** 10.3390/foods9091303

**Published:** 2020-09-16

**Authors:** Judith Ruiz-Martínez, Jorge A. Aguirre-Joya, Romeo Rojas, Antonio Vicente, Miguel A. Aguilar-González, Raúl Rodríguez-Herrera, Olga B. Alvarez-Perez, Cristian Torres-León, Cristóbal N. Aguilar

**Affiliations:** 1Food Research Department. School of Chemistry. Universidad Autónoma de Coahuila, Saltillo, Coahuila 25280, Mexico; judith_ruiz@uadec.edu.mx (J.R.-M.); jorge_aguirre@uadec.edu.mx (J.A.A.-J.); raul.rodriguez@uadec.edu.mx (R.R.-H.); berenice.alvarez.perez@uadec.edu.mx (O.B.A.-P.); ctorresleon@uadec.edu.mx (C.T.-L.); 2Research Center and Development for Food Industries, School of Agronomy, Universidad Autónoma de Nuevo León, General Escobedo NL 66050, Mexico; romeo.rojasmln@uanl.edu.mx; 3CEB-Centre of Biological Engineering, University of Minho, Campus de Gualtar, 4710-057 Braga, Portugal; avicente@deb.uminho.pt; 4Center for Research and Advanced Studies of the National Polytechnic Institute (CINVESTAV-IPN), Unidad Saltillo, Ramos Arizpe, Coahuila 25900, Mexico; miguel.aguilar@cinvestav.edu.mx

**Keywords:** edible coating and films, tomato fruits, *Flourensia cernua*, candelilla, phenolic compounds

## Abstract

The improvement of the postharvest quality of tomato fruits was evaluated using an edible coating functionalized with an *Flourensia cernua* extract evaluating the antifungal, structural, barrier, and optical properties. The formulation and evaluation of an edible coating and its application on tomato was evaluated using a response surface methodology to determine the ideal concentrations of candelilla wax, whey protein, and glycerol. Edible films showed good barrier properties, with water vapor permeability varying from 0.435–0.404 g mm/m^2^ day kPa. The addition o *F. cernua* extract showed significant improvement in the transparency of films. The edible coating applied to tomato reduced weight and firmness loss. The sensory evaluation proved that the product obtained is acceptable for consumers. The edible coating added with *F. cernua* extract was the most effective in inhibiting the growth of pathogenic fungi and the visual appearance at the end of storage confirmed the beneficial effect of the edible coating.

## 1. Introduction

Tomatoes are one of the most commercialized worldwide crops [1]. Mexico is one of the biggest producers of tomatoes in the world and the largest exporter of tomatoes. However, the tomato presents losses close to 50% of the whole production and the losses in the post-harvest of the fruit end up generating environmental problems (odors, water contamination, or pest promotion) and high economic losses [2].

In order to combat all the different factors that cause damage in tomatoes and to preserve their quality, multiple methods have been used. Among the technologies currently used, the use of edible coatings stands out [3]. Edible films and coatings are prepared from naturally occurring renewable sources such as proteins, lipids, alginate, starch, gums, galactomannans, and extracts of various plants rich in bioactive compounds, which present desirable characteristics such as biodegradability, antifungal potential, and good oxygen and vapor barriers [4].

To improve the functional and technological characteristics of films and edible coatings, some materials can be used. Lipids and waxes have been used in formulations to reduce water vapor permeability [5] and formulations with polysaccharides may have low hygroscopicity, constituting poor barriers to water vapor permeability, thus being effective at preserving the quality of food [6]. Whey protein has nutritional and functional properties and represents a by-product of the cheese-making industry. The use of whey protein for edible film and coating formation can improve the nutritional and functional potential of formulations [2].

The most recent discovery in the field of composite edible films refers to the incorporation of bioactive elements such as additives and antioxidant and antimicrobial compounds. Antimicrobial packaging materials can effectively control microbial contamination by inhibiting the growth of spoilage or pathogenic microorganisms on the surface of food [3].

A sustainable source of natural antimicrobial compounds is plant extracts. México has a wide variety of plants, one of which is tarbush (*Fluorensia cernua* D.C.), which is abundant in arid and semiarid regions of Mexico [7]. In vitro studies have shown that *F. cernua* extracts have potent antifungal activity against postharvest pathogens [7,8]. Our research group has recently reported that *F. cernua* have a wide content of compounds with important biological activities; in particular, luteolin 7-O-rutinoside and 6-C-glucosyl-8-C-arabinosyl apigenin are the major metabolites present in tarbush extracts that possess antifungal activities against post-harvest phytopathogens [4].

Even though there are so many reports about edible coatings [3,4,9], more studies are needed, in particular about their formulation and behavior when adding a complex element such as a plant extract and its usefulness in prolonging the shelf life of fruits like tomato. Thus, the objectives of this work were: (a) to formulate edible films of whey protein, candelilla wax, and glycerol with the addition of *F. cernua* extract and (b) to evaluate the application of edible coatings formulated in the shelf life of fresh tomatoes.

## 2. Materials and Methods

### 2.1. Chemicals

Glycerol, reagents, and culture medium for microbiology analysis were obtained from Sigma (St. Louis, MI, USA). Whey protein and candelilla wax of food-grade were supplied by DIA/UAdeC (Departamento de Investigacion en Alimentos/Universidad Autonoma de Coahuila). Potato dextrose agar (PDA) was sourced from Solviosa ^®^ (Jalisco, Mexico).

### 2.2. Obtaining Extracts of F. Cernua

Leaves of tarbush (*F. cernua*) were collected from areas nearby to Saltillo, Coahuila, Mexico. Plant material was dehydrated using a conventional oven (Labnet, International, Inc. Mayfield Ave Edison, NJ, USA) at 60 °C for 2 days. An accurately weighed aliquot of leaves (weight-to-solvent volume ratio 1:10) of dried *F. cernua* was homogenized for 20 min with water. The extract was filtered with a Wathman Num. 1 paper, transferred to glass Petri plates, and then placed in a conventional oven (Labnet, International, Inc.) for 36 h at 60 °C [7]. The dry powder extract was stored in amber bottles at 5 ± 1 °C before use.

### 2.3. Films and Coating Formulation

The filmogenic solution was prepared under a completely randomized design with factorial arrangement, where factors; candelilla wax concentrations of 0.15%, 0.2%, and 0.25% (*w*/*v*); whey protein concentrations of 2%, 2.5%, and 3% (*w*/*v*); and glycerol concentrations of 2%, 2.5%, and 3% (*v*/*v*) were studied. In order to form the films/coatings, whey protein was dissolved in distilled water with constant agitation at 10,000 rpm for 1 min. The mixture was heated at 80 °C for 15 min. Subsequently, candelilla wax and glycerol were added. Finally, the temperature was lowered.

The film was produced from a casting technique according to Torres-León. [10]. Then, 30 mL of the final solution was poured into 9 cm diameter polystyrene plates. The solutions in the plates were dried at 50 °C for 24 h and the film samples that were produced were peeled at an ambient temperature and kept in plastic bags held in desiccators in order to perform thickness and water vapor permeability (WVP) measurements.

In the formulation that presented the best water vapor permeability characteristics, the addition of 500 ppm of *F. cernua* extract was evaluated. In summary, two emulsion-based edible films and coatings were obtained, C: containing whey protein, candelilla wax, and glycerol; and CE: whey protein, candelilla wax, glycerol, and *F. cernua* extract.

### 2.4. Evaluation of Edible Films

#### 2.4.1. Film Thickness

The film thickness was measured with a micrometer (Digimatic Mitutoyo, Japan). Five measurements were taken on different points of each film.

#### 2.4.2. Water Vapor Permeability (WVP)

The WVP was determined gravimetrically according to the method ASTM E96-92 [11] with modifications of Torres-León et al. [10]. The sample cups with the films (100% RH) were weighed every hour for 6 h. Steady-state and uniform water pressure conditions were assumed by keeping the air circulation constant. The WVP was calculated by the following Equation (1).
(1)$$\mathrm{WVP}=\frac{\mathrm{WVTR}\times L}{\Delta P},$$
where WVTR (g/m^2^ day) is the water vapor transmission rate through the film calculated from the slope of the curve divided by the film area; L (mm) is the film thickness; and ΔP (kPa) is the vapor pressure difference across the two sides of the film.

#### 2.4.3. Water Solubility

This parameter was evaluated with the methodology reported by [12]. Pieces of the film of 2 × 3 cm were cut from each film and were dried again in an oven (60 °C). The samples were immersed in 80 mL of distilled water at 30 °C. After 10 min, the samples were dried to a constant weight. The water solubility (%) was calculated by the following Equation (2).
(2)$$\mathrm{Water}\;\mathrm{solubility}\;(\%)\;=\;\frac{\left(\mathrm{Initial}\;\mathrm{weight}-\mathrm{Final}\;\mathrm{weight}\right)}{\left(\mathrm{Initial}\;\mathrm{weight}\right)}\times 100\%.$$

#### 2.4.4. Film Transparency

Film transparency was measured at 600 nm using UV/visible spectrophotometers (Genesys 20, Thermo Scientific™, Waltham, MA, USA) according to the method reported by Zhang et al. [13].

#### 2.4.5. Scanning Electron Microscopy (SEM)

The topography of films was evaluated by SEM (Philips, XL30 ESEM-FEG) with a voltage of 10 kV and a spot with a diameter of 4.5 nm. Before analysis, the samples were sputter-coated with gold (99.9% purity) and were then shredded.

### 2.5. Evaluation of Edible Coating

#### 2.5.1. Fruit grouping and Coating Application

Tomatoes (*Lycopersicon esculentum* Mill. cv Saladette) were obtained from the local market (Saltillo, Coahuila, Mexico). A lot of 180 tomatoes were bought and fruits of uniform size, maturity stage 4, with no physical damage were selected for the experiments. Subsequently, the fruit was washed with distilled water and then surface dried at room temperature.

The coating application was conducted by immersing tomatoes in the following treatments: C (coating without extract), CE (coating with extract), and distilled water (control) for two seconds, then immediately dried under air flow (25 °C) until solidification (approximately 15 min) of the edible coating. As tomatoes are easily perishable, in this study, the coated fruits and control were stored at room temperature of 25 ± 2 °C for 10 days. The fruits were tested each 48 h.

#### 2.5.2. Characterization of Tomatoes

Sensory evaluation was performed with 40 judges aged between 20 and 30 years old. A hedonic scale (1 extremely disliked, 2 dislike, 3 fair, 4 good, and 5 like extremely) was used in the sensory evaluation of fruit coated with the edible material. The following attributes were assessed: color, odor, flavor, and overall appearance [14]. Tomatoes were considered acceptable with a score greater than 3. This evaluation was performed on the 10th day of storage.

The weight loss of tomatoes was evaluated by weighting all samples with a precision balance. The percentage of weight loss was determined by the Equation (3):(3)$$Weight\;loss\;(\%)\;=\;\frac{\left(\mathrm{Initial}\;\mathrm{weight}-\mathrm{Final}\;\mathrm{weight}\right)}{\left(\mathrm{Initial}\;\mathrm{weight}\right)}\times 100\%$$

Firmness loss was carried out in a similar way to the weight loss. Fruit firmness was determined using a universal penetrometer Humboldt (H-1200, Chicaho, IL, USA).

#### 2.5.3. Antifungal Activity In Vitro

Fungal strains *Botrytis cinerea, Colletotrichum gloeosporioides,* and *Fusarim oxysporum* were provided by the Food Research Department of the School of Chemistry (Autonomous University of Coahuila). All fungi were routinely cultured at 28 °C for 7 days on PDA.

Antifungal activity was evaluated following the procedure reported by Rojas et al. [15]. Firstly, PDA medium (previously sterilized) was supplemented with 1 mL of edible coating (C), edible coating with extract (CE), and distilled water (control). Afterward, an explant with a diameter of 0.5 cm of each microorganism was placed in the center of the petri dish. Radial growth was measured at the final time of incubation at 28 ± 2 °C and percent inhibition was calculated with Equation (4).
(4)$$Inhibition\;\left(\%\right)\;=\;\left(\frac{mm\;Growth_{control}-mm\;Growth_{sample}}{mm\;Growth_{control}}\right)\times 100\%$$

#### 2.5.4. Physicochemical Analyses

The pH value was determined using a pH meter (Thermo Orion 420, Beverly, MA, USA). After the homogenization of the samples, pH was measured by direct immersion of the electrode. Titratable acidity was determined using 939.05 AOAC methods [16], specific for fruit derivates, by measuring the amount of 0.1 mol/L NaOH. Results were expressed as % (grams of citric acid equivalent per 100 g) and the soluble solids content was measured with an ATAGO digital refractometer (Tokyo, Japan) and expressed as °Bx.

#### 2.5.5. Color Surface

Tomato surface color was analyzed by measuring Hue angle (h°) with a chromameter (Gretag Macbeth ColorEye XTS USA) at three different points located in the equatorial area.

### 2.6. Statistical Analysis

Statistica 7.0 software (StatSoft, Tulsa, OK, USA) was utilized to analyze the data. The results were analyzed using a completely randomized design 33. Student’s t-test (in the partial characterization of edible films), Analysis of variance (ANOVA), and Tukey’s test were run to determine significant differences between means [10]. The significance level used was 0.05.

## 3. Results and Discussion

### 3.1. Evaluation of Edible Films

WVP measures the diffusion of water molecules through the cross-section of the film and can give an estimation of its barrier properties [17]. WVP is the most extensively studied property of edible films. Figure 1 shows the changes in WVP with the concentration of whey protein, candelilla wax, and glycerol. Low WVP values are ideal in edible films applied to food products since they reduce moisture transfer between the food and the environment [10]. The response surface plot showed that the lowest values of WVP (0.495 ± 0.03 g mm/m^2^ day kPa) are achieved with a combination of 2.5% whey protein, 0.18% of candelilla wax, and 0.6% of glycerol. With this formulation, new films were developed (C). In addition, the incorporation of *F. cernua* extract (CE) was evaluated.

The final results for WVP of the films tested here are shown in Table 1. Films formulated with C and CE did not show a significant difference (*p* < 0.05), which is very interesting since several authors have repopulated that the incorporation of polyphenolic extracts can increase WVP in edible film formulations [10,18,19,20]. The final WVP values determined for C (0.435 ± 0.17 g mm/m^2^ day kPa) and CE (0.404 ± 0.07 g mm/m^2^ day kPa) are lower and consequently, are better than those reported by [21] for films obtained with fish protein (3%, *w*/*v*) and glycerol (30%, *w*/*w*) (7.65 ± 0.03 g mm/m ^2^ day kPa); Pitak & Rakshit [22] for films obtained with chitosan, glycerol, and banana flour (41.66 ± 6.26 g mm/m ^2^ day kPa); and Ramos et al. [23] in films made with whey protein (10.1 g mm/m^2^ day kPa). The addition of the antioxidant extract to the phylogenetic formulation did not significantly affect (*p* < 0.05) the thickness (Table 1). The results determined in this study were similar to those reported by Kim & Ustunol [24] (0.12 mm) and Ramos et al. [23] (0.13 mm) in films prepared with candelilla wax, glycerol, and whey protein.

Water solubility is a parameter that gives an indication of the film’s water affinity, such that higher solubility values indicate a lower resistance to water [25]. As shown in Table 1, the addition of the antioxidant extract contributed to significantly increasing (*p* < 0.05) the solubility in edible films. This behavior has been reported previously in gelatin-based films incorporated with natural antioxidants [26], with the authors mentioning that a high water solubility may be an advantage for some applications. A film with high water solubility can be degraded rapidly and modified easily to improve its physical and chemical properties. The water solubility determined in the present study was similar to that reported by Bourbon et al. [27] in films made with chitosan (42%) and was lower than that reported in sesame protein-based edible film [28] (51–87%).

Transparency is a parameter that can influence the acceptability of films when applied for packaging or food coating; in this analysis, higher values indicate less transparency and more opacity. The addition of antioxidant extract contributed to significantly increasing the transparency of the films (Table 1). The results determined in this study were superior to those previously reported in sesame protein-based edible film [28].

The structural morphology of the films with and without the addition of *F. cernua* extract was studied with the aid of scanning electron microscope (SEM). Figure 2 shows that incorporation of *F. cernua* extracts into the films promoted changes in the film’s morphology extract. CE films presented a structure that was more smooth, continuous, homogeneous, and uniform, which could be due to the entanglement and intermolecular interactions between protein and polyphenols (their ability to form hydrogen bonds or other interactions between the carboxyl groups and the amino acid of proteins) [29]. 

### 3.2. Evaluation of Edible Coatings

Results of the sensory evaluation of tomatoes are plotted in Table 2. All non-coated (control) and coated (C and CE) tomatoes were acceptable for consumption according to panel scores (>3 in a scale of 1—bad to 5—excellent). However, the application of the edible coating with and without antioxidant extract showed a significant difference in all sensory parameters evaluated. The control showed the highest acceptability values. Ochoa et al. [30] explains that these results can be influenced by the judge’s preferences and concepts of taste, color, odor, and overall appearance, so they could have a particular idea of each parameter with respect to the evaluated product.

Weight loss is considered to be the major determinant of storage shelf life and the post-harvest quality of tomatoes [31]. Table 2 shows the total loss of weight of the different treatments. Tomatoes covered with edible coating and antioxidant extract (CE) showed significantly lower weight loss compared to tomatoes treated with cover without extract and with control [29]. These results may be attributed to the formation of a more complex matrix that prevents the migration of water molecules through tomato, reducing water loss [32]. The most weight loss was seen in tomatoes treated with a cover without *F. cernua* extract addition (C). This fact could be due to a formation of a less complex matrix in the surface of the fruit which could have exposed the whey protein and glycerol that because of their hydrophilic character, could have allowed water molecules from the inside to the outside of the tomato, causing weight loss. As mentioned above, the edible film added with antioxidant extract showed a more uniform and organized internal structure than the film without extract [29]. Ruelas-Chacon et al. [33] reported a 15% weight loss in Roma tomato (*Solanum lycoper**sicum* L.) treated with an edible coating of guar gum, values higher than the ones obtained in this study. Lower weight loss in the tomato fruits coated with CE should improve the nutritional value and quality of fruits during shelf-life storage [31].

The evaluation of the loss of firmness at the end of storage showed that tomatoes treatedwith edible coating and antioxidant extract showed significantly less loss of firmness. This behavior is consistent with what is presented in the weight loss. This result could be due to the decreased O_2_ levels and the increase of CO_2_ on the inside of the fruit caused by the edible coating application [34]. 

When these gas levels decrease, the enzymatic activity of membrane enzymes such as pectinesterases and polygalacturonases decreases as well, allowing a minor firmness loss in tomato. Moreno et al. [35] mentions that the active ingredients have the ability to maintain firmness by inhibiting enzymes responsible for deteriorating the firmness of tomatoes.

#### 3.2.1. Antifungal Activity In Vitro

As shown in Table 2, the treatment with CE presented the highest inhibition values against pathogenic fungi *B. cinerea* (35%), *C. gloeosporioides* (39%), and *F. oxysporum* (6%) and these percentages of inhibition were significant with respect to C and control. This could be due to the combination of *F. cernua* extract with the other components of the edible coating, such as candelilla wax, resulting in a better antimicrobial capacity. Antifungal activity of *F. cernua* can be referred to as the presence of bioactive compounds, such as gallic acid, luteolin 7-O-rutinoside, and apigenin galactoside arabinoside [7]. These results confirm the in vitro antifungal activity, previously reported by Jasso-Rodríguez et al. [8] and De León-Zapata et al. [7], in extracts of *F. cernua*. The application of edible films and coating can contribute to delaying the microbiological damage of fruit. Additionally, the addition of polyphenolic compounds considerably improves the antifungal potential of the formulations [32,36].

#### 3.2.2. Physicochemical Analyses

The pH values of all treatments increased with the storage time and were not statistically different (Figure 3). In general, there is an approximate pH increase of 4.1 to 4.3. These results agree with those reported by Dávila-Aviña et al. [6], which observed an increase of pH from 4.0 to 4.6. On the final day of storage, CE presents the lowest pH values, followed by control and treatment C. This behavior and the values determined in this study were similar to those reported on day 9 of tomato (*S. lycopersicum* L.) treated with a nanolaminate with extract (4.15), Control (4.26), and nanolaminate without extract (4.37) [37]. In general, titratable acidity values decreased in all treatments along the storage period (Figure 3) and this reduction is related to organic acid degradation, such as citric and malic acid (major in tomato), due to the respiration process of the fruit [38].

During the time course of storage, a gradual increase in soluble solids concentration was observed in all tested samples (Figure 3). This behavior is in line with what is reported in the literature, since during the fruit ripening process, the level of soluble sugars increases as a result of the activation of many genes such as vacuolar invertase and sucrose synthase [39]. Initially, the values of soluble solids were constant, but after the second day of storage, the values increased, without presenting a significant difference between the treatments. According to Beckles [40], soluble solids content is a parameter closely related to the titrable acidity percentage in the fruit, which means that when acidity percentage decreases, the total soluble solids increases. This situation is presented due to the fact that during the respiration process of the plant, organic acids are used as substrates and are hydrolized to sugars, increasing the Brix.

Hue angle values decreased in all treatments throughout tomato storage. In general, the decrease in hue angle values indicates that tomatoes changed from an initial orange color (48 h°) to red (40–42 h°). This decreasing in Hue angle was lower in tomatoes treated with C and CE with respect to the control, showing a significant difference at the end of storage. During the ripening process, the green pigment of fruit, corresponding to chlorophyll, is degraded and carotenoids are accumulated (lycopene principally) [34]. This decreasing in the color change of coated tomatoes can be caused by an increase in CO_2_ levels, causing a decrease in ethylene production of the fruit, which also delays the natural ripening process of tomato, avoiding the climacteric response induction and therefore, delaying color-changing [41]. The results of this study are consistent with what was reported by Chaturvedi et al. [42] in tomatoes (*S. lycopersicum*) treated with pectin-corn flour coatings. The authors mention that the delayed effect can be correlated with the results obtained in the firmness loss (Table 2) and these parameters may have been influenced by ethylene concentration.

Visual evaluation of tomatoes (at the 10th day of storage) confirmed the favorable effect of the edible coating. All fruits presented changes in their external visual appearances (Figure 4), but the tomatoes with C (Figure 4B) and CE (Figure 4C) suffered a minor change compared with the control. These results are consistent with what was reported in fruits covered with films formulated with candelilla wax [43,44]. De León-Zapata et al. [45] reported that apples covered with candelilla wax and fermented extract of tarbush showed better appearance at the end of storage (8 weeks) compared to apples treated with wax without extract or the control. Similar results have been reported on the application of edible coating based waxes with other natural active compounds [46,47].

This study shows that the application of an edible coating based on candelilla wax and functionalized with *F. cernua* extract extends the shelf-life of tomato fruit. This research contributes to generating important social, economic, and environmental benefits. Tomato growers can improve their marketing options and reduce post-harvest losses. Additionally, as Candelilla (*Euphorbia antisyphilitica*) and *F. cernua* are endemic plants of the arid and semiarid regions of Mexico [48], the development of postharvest technologies based on these plants can contribute to the generation of added value for local communities. New research is needed to study sustainable extraction methods for candelilla wax and *F. cernua* phytochemicals. Technology transfer projects with rural producers or companies should also be considered.

## 4. Conclusions

An oleo-proteic edible film can be successfully formed combining candelilla wax, whey protein, and glycerol. The addition of the *F. cernua* extract to the films did not significantly affect permeability and thickness. Films with extract presented a structure that was more smooth, continuous, homogeneous, and uniform. The edible coating applied to tomato reduced weight and firmness loss in storage for 10 days at 25 °C. The edible coating added with *F. cernua* extract was the most effective at inhibiting the growth of pathogenic fungi and the visual appearance at the end of storage confirmed the beneficial effect of the edible coating. Therefore, our results indicated that the addition of bioactive extract from *F. Cernua* exerts superior effects than the individual coatings and can contribute to expanding the shelf-life of tomato fruits.

## Figures and Tables

**Figure 1 foods-09-01303-f001:**
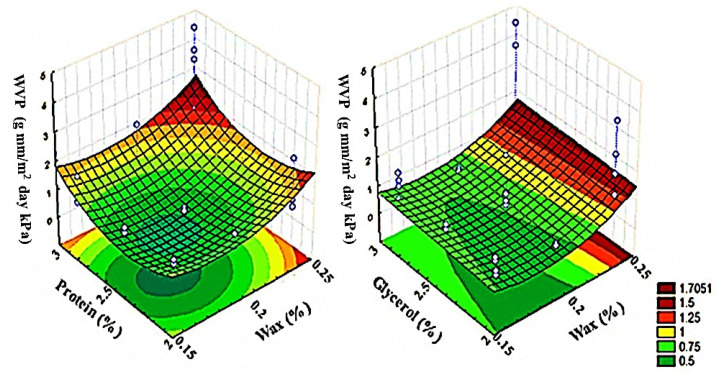
Response surface graph of Water Vapor Permeability (WVP) in films as a function of protein, wax, and glycerol concentrations.

**Figure 2 foods-09-01303-f002:**
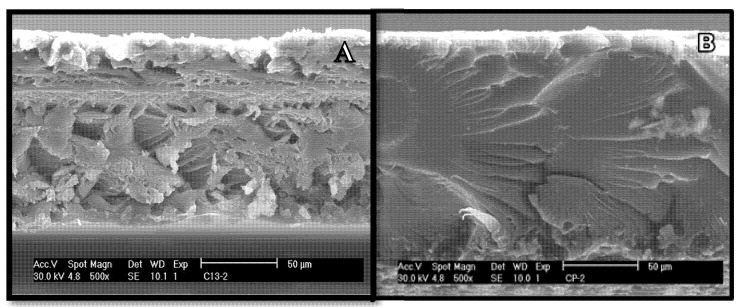
Scanning electron microscopy (SEM). (**A**) Edible film, (**B**) Edible film with *F. cernua* extract.

**Figure 3 foods-09-01303-f003:**
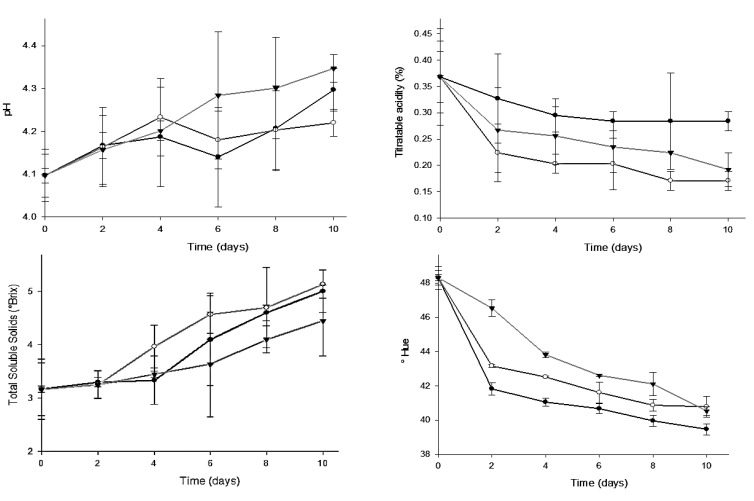
pH values, titratable acidity, total soluble solids content, and color surface (Hue angle) of tomato fruits covered with edible coating (Triangle) or edible coating and *F. cernua* extract (White dot) or distilled water as a control (Black dot).

**Figure 4 foods-09-01303-f004:**
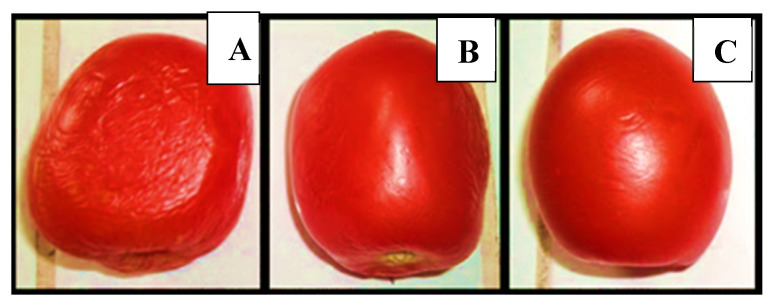
Visual aspect of tomatoes on the tenth day of storage (25 °C). (**A**) Uncoated tomato samples (Control), (**B**) coated samples, (**C**) coated samples with *F. cernua* extract.

**Table 1 foods-09-01303-t001:** Partial characterization of edible films.

Parameter	C	CE
WVP (g mm/m^2^ day kPa)	0.435 ± 0.17 ^a^	0.404 ± 0.07 ^a^
Thickness (mm)	0.10 ± 0.01 ^a^	0.09 ± 0.01 ^a^
Solubility (%)	44.52 ± 0.35 ^a^	48.62 ± 0.62 ^b^
Transparency (A600/mm)	13.17 ± 0.44 ^b^	10.36 ± 0.50 ^a^

Values reported are the means ± standard deviations. Means with different letters are significantly different (*p* < 0.05).

**Table 2 foods-09-01303-t002:** Sensory evaluation, weight loss, firmness loss, and antifungal activity of edible coating.

Evaluation		Control	C	CE
Sensory evaluation	Overall appearance	4.4 ^a^	3.6 ^b^	3.5 ^b^
	Color	4.3 ^a^	3.9 ^b^	3.4 ^c^
	Odor	3.8 ^a^	3.6 ^b^	3.6 ^b^
	Flavor	4.1 ^a^	3.3 ^b^	3.3 ^b^
Weight loss (%)		9.0 ^b^	13.0 ^c^	7.3 ^a^
Firmness loss (%)		30.0 ^c^	19.4 ^b^	14.60 ^a^
Inhibition (%)	*B. cinerea*	0.0 ^c^	23.0 ^b^	35.0 ^a^
	*C. gloeosporioides*	0.0 ^c^	28.0 ^b^	39.0 ^a^
	*F. oxysporum*	0.0 ^c^	3.0 ^b^	6.0 ^a^

Means with different letters are significantly different (*p* < 0.05).

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
