# Peer review of "Candelilla Wax Edible Coating with Flourensia cernua Bioactives to Prolong the Quality of Tomato Fruits"

_foods, 2020, doi:10.3390/foods9091303_

Round 1
Reviewer 1 Report
Please find comments in the text.
Manuscript is well prepared.

Author Response
1. Line 75. The explanation is not clear. The extract was dried and added in a powder form? How it was stored or conditioned? Answer: The reviewer's observation was revised. Storage details were added Line 77. The dry powder extract was stored in amber bottles at 5 ± 1 °C before use. 2. Line 85. There is no information about the pH Answer: The pH was not a variable in the formulation of the coatings. This parameter was evaluated in the postharvest of the fruits. 3. Line 110. Why this pemperature was used? Usually it is 20 or 25 degrees Answer: The temperature was selected according to the methodology reported by Romero-Bastida et al., 2005 (doi: 10.1016 / j.carbpol.2005.01.004). The authors report that room temperature should be used. They used approximately 25 ° C, in our study the ambient temperature was 30 °C (± 5 °C). 4. Line 169. There is lack of explanation regarding WVP. It should be compared wiyh film microsturucture since wax contaning filmw were highest WVP which is probably conected with more porous structure Answer: WVP is the result of the combined formulation of the three constituents wax, protein, and glycerol at different levels. The complexity of the microstructures formed is very high, this makes the comparison of each component very difficult, these details are inferred in the decrease in permeability evaluated by the statistical design. 5. Line 172. This is not the same concentration indicated in Method section The evaluator's observation is very important to us. These values were obtained by the mathematical modeling of the variables (The response surface plot). This statistical design provides the best process conditions for the evaluated levels, the central values of protein and wax were close to the optimal points, however, the lower the glycerol concentration, the better the permeability. These results are very useful for future studies to determine their study conditions, the findings are also important to study perspectives for our research group. 6. Line 303. Some remarks about edible films should be added Answer: The reviewer's comment is very important to improve the manuscript. The conclusion was modified: Line 303-308. The addition of the F. cernua extract to the films did not significantly affect permeability and thickness. Films with extract presented a structure more smooth, continuous, homogeneous, and uniform.
Reviewer 2 Report
In the manuscript, the characteristics of the composite film with whey protein, candelilla wax and glycerol with the addition of extracts of F. cernua on the post-harvest quality of tomatoes were investigated. The research results are interesting, but two methodological errors were made. The first one concerns the determination of the concentration of glycerol in the composite film. The second error concerns the use of the mistakenly method to evaluate the anti-mold activity of the film.
Detailed comments:
- Lines 140 and 249-255: Latin names of mold species and plant should be in italics.
- Lines 130-134: When was the sensory analysis of tomatoes done? Immediately after coating or after 10 days of storage?
- Lines 139-147: Adding solution C or CE to the PDA medium and inoculating the mold is a method to test the anti-mold activity of all components of the solution, not the film. It was necessary to inoculate the mold on the surface of the PDA medium and put a disc of the composite film on it. Zones of mold growth inhibition are measured.
- Line 172 (Fig.1): 0.6% glycerol is out of scale. Glycerol was tested over the range of 2, 2.5 and 3% (line 81 and Fig. 1), not lower. The conclusion is not substantiated by the results of the experiment.
- Lines 212-213: SEM photos have a different scale, but should be of the same scale. The scale in Fig. 2a is smaller than the scale in Fig. 2B, so it is difficult to objectively compare the structure of the two films.
- Line 223: Fig. 3 didn’t shows the total loss of weight of the different treatments.
- Line 237 (Table 2): There are no scores of overall appearance of tomatoes (line 133). Which day of storage did test tomatoes? Section 2.5.1 states that coated and uncoated tomatoes were stored for 10 days and tested every 48 h. Why are these results not shown in Table 2 ?
- Lines 247-258: The film solution was tested, not the composite film against mold. In this experiment only shown the antimold activity of the cernua extract.
Author Response
In the manuscript, the characteristics of the composite film with whey protein, candelilla wax and glycerol with the addition of extracts of F. cernua on the post-harvest quality of tomatoes were investigated. The research results are interesting, but two methodological errors were made. The first one concerns the determination of the concentration of glycerol in the composite film. The second error concerns the use of the mistaken method to evaluate the anti-mold activity of the film.
- Lines 140 and 249-255: Latin names of mold species and plant should be in italics.
Answer:
The scientific names of the species used in the study were corrected:
Line 90. F. cernua
Line 92. F. cernua
Line 140. Botrytis cinerea, Colletotrichum gloeosporioides, and Fusarim oxysporum
Line 173. F. cernua
Line 207. F. cernua
Line 213. F. cernua
Line 228. F. cernua
Line 249. B. cinerea , C. gloeosporioides and F. oxysporum
Line 251. F. cernua
Line 271. F. cernua
Line 303. F. cernua
- Lines 130-134: When was the sensory analysis of tomatoes done? Immediately after coating or after 10 days of storage?
Answer:
The sensory analysis was carried on the 10th day of storage according to the methodology described by Fakhouri et al., 2015.
- Lines 139-147: Adding solution C or CE to the PDA medium and inoculating the mold is a method to test the anti-mold activity of all components of the solution, not the film. It was necessary to inoculate the mold on the surface of the PDA medium and put a disc of the composite film on it. Zones of mold growth inhibition are measured.
Answer:
The evaluator's suggestion is very important to our research group. In the present investigation, we tried to evaluate the antifungal activity of the solution applied to tomatoes, the technique used is close to what happens in the fruit coated with a solution (the technique is in the edible coatings section). The coating of the fruit may vary from that obtained in the Petri dish. Therefore, we decided not to evaluate the film's disk and to evaluate the entire solution. The technique suggested by the evaluator is very valuable, this contribution will be considered for future investigations of our research.
- Line 172 (Fig.1): 0.6% glycerol is out of scale. Glycerol was tested over the range of 2, 2.5 and 3% (line 81 and Fig. 1), not lower. The conclusion is not substantiated by the results of the experiment.
Answer:
This observation is very important to improve the quality of the document. The values that gave the lowest water vapor permeability were chosen with mathematical modeling.
- Lines 212-213: SEM photos have a different scale, but should be of the same scale. The scale in Fig. 2a is smaller than the scale in Fig. 2B, so it is difficult to objectively compare the structure of the two films.
Answer:
The suggestion of the evaluator is very important to improve the manuscript. The scale of the figure was corrected, figure 2a was changed. The scale of the photos is 500x.
- Line 223: Fig. 3 didn’t shows the total loss of weight of the different treatments.
Answer:
The evaluator's observation is very important. Figure 3 does not show the total weight loss, that information was presented in Table 3. The correction was made in the document.
Line. 223. Table 3 shows the total loss of weight of the different treatments.
- Line 237 (Table 2): There are no scores of overall appearance of tomatoes (line 133). Which day of storage did test tomatoes? Section 2.5.1 states that coated and uncoated tomatoes were stored for 10 days and tested every 48 h. Why are these results not shown in Table 2 ?
Answer:
The results of the overall appearance of tomatoes are reported in table 2. The tomatoes were evaluated at the end of storage, this information was added to the document.
Line 134. This evaluation was performed on the 10th of storage.
The physicochemical tests were evaluated every 48 hours, the sensory analysis was at the end of storage.
- Lines 247-258: The film solution was tested, not the composite film against mold. In this experiment only shown the antimold activity of the cernua extract.
Answer:
In the experiment, the antifungal activity of the coating was evaluated with and without the addition of the bioactive extract. the film was not evaluated since in the practice on the fruit the coating solution was evaluated. We wanted to know if under these conditions the addition of extract influences the growth of molds.

Reviewer 3 Report
Dear authors,
the current manuscript describes the application of edible coatings on fresh tomatoes, and differences were described taking into account controls and samples with a novel natural antimicrobial extract.
The manuscript shows an adequate design of methods for the proposed work. The English of the manuscript is accurate, but please, make another revision on English to cover the minus errors and the miss-spelling.
Please find small notes throughout all the manuscript:
Line 71: F. cernua should come in extense, in the title.
Line 72-77: F. cernua, should come in italic.
Line 85: The reference should have the name and not number.
Line 90/92: F. cernua, should come in italic.
Line 99/108: The reference should have the name and not number.
Line 109: Correct please the following sentence: … in an oven at 60 °C)
Line 114: The reference should have the name and not number.
Line 120: Lycopersicon esculentum, should come in italic.
Line 139-140: In vitro and the names of fungal strains, should come in italic.
Line 143: The reference should have the name and not number.
Line 207-208: F. cernua, should come in italic.
Line 209-211: Please find a bibliographic justification for this assumption.
Line 228: F. cernua, should come in italic.
Line 227-231: Do you have bibliographic support on that assumption?
Line 233/265/292: Solanum lycopersicum L. , should come in italic.
Line 234: Change to: guar gum.
Line 247: In vitro and the names of fungal strains, should come in italic.
Line 252/255/303/305: F. cernua, should come in italic.
Line 254: In vitro, should come in italic.
Figure 3: F. cernua, should come in italic.
The results and discussion are well written. But lacks, bibliographic references to support made assumption, please find accurate and adequate bibliographic support.
The tables and figures are enough for the results obtained and discussed results. But, Figure 3, should be improved, the symbols of each curve should be bigger, to be noticed easily. Table 2, should be reformulated, some parts are in bold and others are not, could the authors standardize.
The conclusion section must be improved, the information should be more incorporative of all obtained results and should be conclusive.
Author Response
The current manuscript describes the application of edible coatings on fresh tomatoes, and differences were described taking into account controls and samples with a novel natural antimicrobial extract.
The manuscript shows an adequate design of methods for the proposed work. The English of the manuscript is accurate, but please, make another revision on English to cover the minus errors and the miss-spelling.
Please find small notes throughout all the manuscript:
- Line 71: F. cernua should come in extense, in the title.
Answer:
The title has the name of the plant in extense
Line 2-3. Candelilla wax edible coating with Flourensia cernua bioactives to prolong the quality of tomato fruits
- Line 72-77: F. cernua, should come in italic.
Answer:
The scientific names of the species used in the study were corrected:
Line 72. F. cernua
- Line 85: The reference should have the name and not number.
Answer:
Evaluator observation is very important. The reference must have the number. Also, we add the name of the authors.
Line 85. The film was produced from casting technique (30 mL) according to Torres-León. [10], which were dispersed in polystyrene plates (9 cm) and dried at 50 °C for 16 h.
- Line 90/92: F. cernua, should come in italic.
Answer:
The scientific names of the species used in the study were corrected:
Line 90-92. In the formulation that presented the best water vapor permeability characteristics, the addition of 500 ppm of F. cernua extract was evaluated. In summary, two emulsion-based edible films and coatings were obtained, C: containing whey protein, candelilla wax, and glycerol, and CE: whey protein, candelilla wax, glycerol, and F. cernua extract.
- Line 99/108: The reference should have the name and not number.
Answer:
According to the journal, the references must be with numbers, however, we add the name of the author.
Line 98-99. The WVP was determined gravimetrically according to the method ASTM E96-92 [11] with modifications of Torres-León et al. [10].
- Line 109: Correct please the following sentence: … in an oven at 60 °C)
Answer:
The sentence was corrected
Line 108-109. Pieces of the film of 2 x 3 cm were cut from each film and were dried again in an oven (60 ° C)
- Line 114: The reference should have the name and not number.
Answer:
The evaluator's suggestion was answered.
Line 114. Film transparency was measured at 600 nm using UV/visible spectrophotometers (Genesys 20, USA) according to the method reported by Zhang et al. [13].
- Line 120: Lycopersicon esculentum, should come in italic.
Answer:
The scientific names of the species used in the study were corrected.
Line 120. Tomatoes (Lycopersicon esculentum Mill. cv Saladette) were obtained from the local market (Saltillo, Coahuila, Mexico).
- Line 139-140: In vitro and the names of fungal strains, should come in italic.
Answer:
The corrections were made.
Line 139-142. Antifungal activity in vitro
Fungal strains Botrytis cinerea, Colletotrichum gloeosporioides, and Fusarim oxysporum were provided by the Food Research Department of the School of Chemistry (Autonomous University of Coahuila). All fungi were routinely cultured at 28 °C for 7 d on PDA.
- Line 143: The reference should have the name and not number.
Answer:
The evaluator's suggestion was answered.
Line 143. Antifungal activity was evaluated following the procedure reported by Rojas et al. [15].
- Line 207-208: F. cernua, should come in italic.
Answer:
The scientific names of the species used in the study were corrected:
Line 207. F. cernua
The structural morphology of the films with and without the addition of F. cernua extract was studied with the aid of scanning electron microscope (SEM).
- Line 209-211: Please find a bibliographic justification for this assumption.
Answer:
New bibliography was added to the document.
Line 209-213. CE films presented a structure more smooth, continuous, homogeneous and uniform, which could be due to the entanglement and intermolecular interactions between protein and polyphenols (their ability to form hydrogen bonds or other interactions between the carboxyl groups and the amino acid of proteins) [29].
The new reference was added
Line 399. Gariani, S.; Choulitoudi, E.; Oreopoulou, V. Edible and active films and coatings as carriers of natural antioxidants for lipid food. Trends in Food Science & Technology 2017, 68, 70-82, doi:10.1016/j.tifs.2017.08.009
- Line 228: F. cernua, should come in italic.
Answer:
The scientific names of the species used in the study were corrected:
Line 228. F. cernua
- Line 227-231: Do you have bibliographic support on that assumption?
Answer:
The bibliographic reference was added.
Line 227. compared to tomatoes treated with cover without extract and with control [29].
The new reference was added
Line 399. Gariani, S.; Choulitoudi, E.; Oreopoulou, V. Edible and active films and coatings as carriers of natural antioxidants for lipid food. Trends in Food Science & Technology 2017, 68, 70-82, doi:10.1016/j.tifs.2017.08.009
- Line 233/265/292: Solanum lycopersicum L. , should come in italic.
Answer:
The scientific names of the species used in the study were corrected:
Line 235. Chacon et al. [32], reported a 15% weight loss in Roma tomato (Solanum lycopersicum L.)
Line 267. (S. lycopersicum L.) treated with a nanolaminate with extract (4.15), Control (4.26), and nanolaminate without extract (4.37) [35].
Line 295. Chaturvedi et al. [40] in tomatoes (S. lycopersicum) treated with pectin-corn flour coatings
- Line 234: Change to: guar gum.
Answer:
The correction was made.
Line 236. an edible coating of guar gum
- Line 247: In vitro and the names of fungal strains, should come in italic.
Answer:
The scientific names of the species used in the study were corrected:
Line 249-252. Antifungal activity in vitro
As shown in Table 2, the treatment with CE, presented the highest inhibition values against pathogenic fungi B. cinerea (35%), C. gloeosporioides (39 %) and F. oxysporum (6 %),
- Line 252/255/303/305: F. cernua, should come in italic.
Answer:
The scientific names of the species used in the study were corrected:
Line 253. combination of F. cernua extract with the other components
Line 254. Antifungal activity of F. cernua can be referred to
Line 302. (Control), (B) coated samples, (C) coated samples with F. cernua extract.
Line 305. The addition of the F. cernua extract to the films did not significantly affect permeability
- Line 254: In vitro, should come in italic.
Answer:
The correction was made.
Line 256. These results confirm the in vitro antifungal activity
- Figure 3: F. cernua, should come in italic.
Answer:
The correction was made.
Line 272. Figure 3. pH values, titratable acidity, total soluble solids content and color surface (Hue angle) of tomatoes fruits covered with the edible coating (Triangle) or edible coating and F. cernua extract (White dot) or distilled water as control (Black dot).
- The results and discussion are well written. But lacks, bibliographic references to support made assumption, please find accurate and adequate bibliographic support.
Answer:
The suggestion of the evaluator is very important to improve the quality of the manuscript. The bibliographic references were reviewed in the results and discussion section. Additionally, references [29] (Line 212), [32] (Line 229) and [32,36] (Line 260) were added.
New literature was added in the references
Line 401: 29. Gariani, S.; Choulitoudi, E.; Oreopoulou, V. Edible and active films and coatings as carriers of natural antioxidants for lipid food. Trends in Food Science & Technology 2017, 68, 70-82, doi:10.1016/j.tifs.2017.08.009
Line 409: 32. Pobiega, K.; Igielska, M.; Włodarczyk, P.; Gniewosz, M. Prolonging the Shelf Life of Cherry Tomatoes by Pullulan Coating with Ethanol Extract of Propolis During Refrigerated Storage. Food and Bioprocess Technology 2020, 13, 1447-1461, doi.org/10.1007/s11947-020-02499-6.
Line 423: 36. Pobiega, K.; Igielska, M.; Włodarczyk, P.; Gniewosz, M. The use of pullulan coatings with propolis extract to extend the shelf life of blueberry (Vaccinium corymbosum) fruit. International Journal of Food Science & Technology 2020, doi: 10.1111/ijfs.14753.
- The tables and figures are enough for the results obtained and discussed results. But, Figure 3, should be improved, the symbols of each curve should be bigger, to be noticed easily. Table 2, should be reformulated, some parts are in bold and others are not, could the authors standardize.
Answer:
The suggestion of the evaluator is very important to improve the quality of the manuscript.
The sharpness of figure 3 was improved so that the symbols can be differentiated. Table 2 was standardized and improved.
- The conclusion section must be improved, the information should be more incorporative of all obtained results and should be conclusive.
Answer:
The conclusions section was improved.
Line 304. An oleo-proteic edible film can be successfully formed combining candelilla wax, whey protein, and glycerol. The addition of the F. cernua extract to the films did not significantly affect permeability and thickness. Films with extract presented a structure more smooth, continuous, homogeneous, and uniform. The edible coating applied to tomato reduced weight and firmness loss in storage for 10 days at 25 ° C. The edible coating added with F. cernua extract, was the most effective to inhibit the growth of pathogenic fungi, the visual appearance at the end of storage confirmed the beneficial effect of the edible coating. Therefore, our results indicated that the addition of bioactive extract from F. Cernua exerts more superior effects than the individual coatings and can contribute to expanding the shelf-life of tomato fruits.

Reviewer 4 Report
The manuscript on tomato coatings presented for review is very interesting. The manuscript brings new insight into edible coatings through the use of candelilla wax and Flourensia cernua extracts. The manuscript fits into the subject of the journal for which it was submitted for review.
I would like to ask you to clarify some details:
Some names are not italicized throughout the manuscript, e.g. line 90, 92, 120, 140
In line 130 - were the judges qualified or not?
In line 48-52 should add more literature, for example DOI: 10.1111/ijfs.14753, DOI: 10.1007/s11947-020-02487-w
Author Response
The manuscript on tomato coatings presented for review is very interesting. The manuscript brings new insight into edible coatings through the use of candelilla wax and Flourensia cernua extracts. The manuscript fits into the subject of the journal for which it was submitted for review.
I would like to ask you to clarify some details:
- Some names are not italicized throughout the manuscript, e.g. line 90, 92, 120, 140
Answer:
The scientific names of the species used in the study were corrected:
Line 90. F. cernua
Line 92. F. cernua
Line 120. Lycopersicon esculentum
Line 140. Botrytis cinerea, Colletotrichum gloeosporioides, and Fusarim oxysporum
Line 173. F. cernua
Line 207. F. cernua
Line 213. F. cernua
Line 228. F. cernua
Line 249. B. cinerea, C. gloeosporioides and F. oxysporum
Line 251. F. cernua
Line 271. F. cernua
Line 303. F. cernua
- In line 130 - were the judges qualified or not?
Answer:
The judges were not qualified.
- In line 48-52 should add more literature, for example DOI: 10.1111/ijfs.14753, DOI: 10.1007/s11947-020-02487-w
Answer:
New references were added to improve the manuscript
Line 212. [29].
Line 229. [32].
Line 260. [32,36].
New literature was added in the references
Line 401: 29. Gariani, S.; Choulitoudi, E.; Oreopoulou, V. Edible and active films and coatings as carriers of natural antioxidants for lipid food. Trends in Food Science & Technology 2017, 68, 70-82, doi:10.1016/j.tifs.2017.08.009
Line 409: 32. Pobiega, K.; Igielska, M.; Włodarczyk, P.; Gniewosz, M. Prolonging the Shelf Life of Cherry Tomatoes by Pullulan Coating with Ethanol Extract of Propolis During Refrigerated Storage. Food and Bioprocess Technology 2020, 13, 1447-1461, doi.org/10.1007/s11947-020-02499-6.
Line 423: 36. Pobiega, K.; Igielska, M.; Włodarczyk, P.; Gniewosz, M. The use of pullulan coatings with propolis extract to extend the shelf life of blueberry (Vaccinium corymbosum) fruit. International Journal of Food Science & Technology 2020, doi: 10.1111/ijfs.14753.

Round 2
Reviewer 2 Report
Two comments:
Line 291: The sentence was not corrected. Table 3 does not in the manuscript. The correct sentence is: Table 2 shows the total loss of weight of the different treatments.
Lines 646: Please, correct the authors of the publication: Pobiega, K., Przybył, J.L., Żubernik, J., Gniewosz, M. Prolonging the Shelf Life of Cherry Tomatoes by Pullulan Coating with Ethanol Extract of Propolis During Refrigerated Storage. Food and Bioprocess Technology 2020, 13, 1447-1461, doi.org/10.1007/s11947-020-02499-6
Author Response
Reviewer #2:
- Line 291: The sentence was not corrected. Table 3 does not in the manuscript. The correct sentence is: Table 2 shows the total loss of weight of the different treatments.
Answer:
The evaluator's comment is very important to improve the quality of the manuscript. This was done.
Line 224. Table 2 shows the total loss of weight of the different treatments.
- Lines 646: Please, correct the authors of the publication: Pobiega, K., Przybył, J.L., Żubernik, J., Gniewosz, M. Prolonging the Shelf Life of Cherry Tomatoes by Pullulan Coating with Ethanol Extract of Propolis During Refrigerated Storage. Food and Bioprocess Technology 2020, 13, 1447-1461, doi.org/10.1007/s11947-020-02499-6
Answer:
The correction was made.
Line 408. Pobiega, K., Przybył, J.L., Żubernik, J., Gniewosz, M. Prolonging the Shelf Life of Cherry Tomatoes by Pullulan Coating with Ethanol Extract of Propolis During Refrigerated Storage. Food and Bioprocess Technology 2020, 13, 1447-1461, doi.org/10.1007/s11947-020-02499-6
